

# Single-nucleotide polymorphisms of uracil-processing genes affect the occurrence and the onset of recurrent depressive disorder

Piotr Czarny[1], Paulina Wigner[2], Justyna Strycharz[1], Cezary Watala[3], Ewa Swiderska[1], Ewelina Synowiec[2], Piotr Galecki[4], Monika Talarowska[4], Janusz Szemraj[1], Kuan-Pin Su[5] and Tomasz Sliwinski[2]

[1] Department of Medical Biochemistry, Medical University of Lodz, Lodz, Poland
[2] Laboratory of Medical Genetics, Faculty of Biology and Environmental Protection, University of Lodz, Lodz, Poland
[3] Department of Haemostatic Disorders, Medical University of Lodz, Lodz, Poland
[4] Department of Adult Psychiatry, Medical University of Lodz, Lodz, Poland
[5] Department of Psychiatry and Mind-Body Interface Laboratory (MBI-Lab), China Medical University Hospital, Taichung, Taiwan

Corresponding author
Tomasz Sliwinski,
tomasz.sliwinski@biol.uni.lodz.pl,
tomsliw@biol.uni.lodz.pl

## ABSTRACT

Depressive disorders (DD) are known to be associated with increased DNA damage, the impairment of DNA damage repair, and the presence of single-nucleotide polymorphisms (SNPs) in DNA damage repair genes. Some indirect evidence also suggests that uracil metabolism may be disrupted in depressed patients. Therefore, the current study genotypes three SNPs localized in genes encoding uracil-processing proteins: two glycosylases, i.e., *UNG* g.7245G>C (rs34259), *SMUG1* c.-31A>G (rs3087404), and dUTPase, i.e., *DUT* g.48638795G>T (rs4775748). The polymorphisms were analyzed in 585 DNA samples (282 cases and 303 controls) using TaqMan probes. The G/G genotype and G allele of *UNG* polymorphism decreased the risk of depression, while the G/C genotype and C allele of the same SNP increased it. It was also found that G/G carriers had their first episode significantly later than the heterozygotes. Although there was no association between the occurrence of depression and the *SMUG1* SNP, a significant difference was found between the homozygotes regarding the onset of DD. In conclusion, the SNPs localized in the uracil-processing genes may modulate the occurrence and the onset of depression, which further supports the hypothesis that impairment of DNA damage repair, especially base-excision repair, may play an important role in the pathogenesis of the disease.

## INTRODUCTION

By 2020, depression (depressive disorders—DD, including recurrent ones–rDD) is predicted to be the most profound economic and social burden next to ischemic heart disease (*Murray & Lopez, 1997*; *Greden, 2001*). Unfortunately, despite extensive research, its pathogenesis, etiology, diagnosis and treatment remain unclear

(*Gruenberg, Goldstein & Pincus, 2005*). No cellular or molecular biomarkers exist, and so clinical diagnosis is based only on observational manifestations. In addition, one-third of patients suffers from treatment-resistant depression (*Brown, Harris & Hepworth, 1994*; *Parker, 2000*; *Gruenberg, Goldstein & Pincus, 2005*; *Akiskal & Lewis, 2005*; *Starkstein et al., 2005*; *Iwata, Ota & Duman, 2013*; *Rodrigues, Petersen & Perry, 2014*; *Chang et al., 2014*).

Some factors are well known to play a crucial role in depression. Inflammation, and elevated oxidative and nitrosative stress, have recently attracted much attention (*Pasco et al., 2010*; *Gardner & Boles, 2011*; *Alcocer-Gómez et al., 2014*; *Anderson & Maes, 2014*; *Lu, Leung & Su, 2013*). These kinds of stress may induce damage to biomolecules, including nucleic acids. Accordingly, an elevated level of 8-oxoguanine (8-oxoG), a marker of oxidative DNA damage, has been found in the serum, peripheral blood mononuclear cells (PBMCs) and urine of patients suffering from clinical depression and/or depression coexisting with other non-psychiatric diseases (*Irie et al., 2001*; *Irie et al., 2003*; *Forlenza & Miller, 2006*; *Maes et al., 2009*; *Wei et al., 2009*; *Kupper et al., 2009*; *Black et al., 2015*; *Lindqvist et al., 2017*).

Our team revealed that the PBMCs isolated from depressed patients demonstrated slower DNA damage repair efficiency (DRE) after exposure to hydrogen peroxide than controls (*Czarny et al., 2015a*). Moreover, we found that the presence of single-nucleotide polymorphisms (SNPs) encoding proteins involved in base excision repair (BER), a primary pathway used to ameliorate oxidative DNA damage, also appear to affect the occurrence and onset of depression (*Czarny et al., 2015b*; *Czarny et al., 2016*). Although a genotype-phenotype analysis indicated that these SNPs did not generally influence the levels of DNA damage in the patients' cells, some may still have an impact on DRE (*Czarny et al., 2017a*). This leads us to believe that the increased DNA damage observed in depression may be caused not only by oxidative or/and nitrosative stress, but also by the impairment of DNA damage repair pathways, particularly BER (*Czarny et al., 2017b*).

The BER pathway is also responsible for the removal of uracil and its derivatives from DNA (*Barnes & Lindahl, 2004*; *Visnes et al., 2009*). Uracil DNA glycosylases (UDGs) initiate repair by recognizing and excising uracil. Several enzymes in the UDG superfamily have been identified, including uracil-N glycosylase (UNG), single-strand-specific monofunctional uracil DNA glycosylase 1 (SMUG1), methyl-binding domain glycosylase 4 (MBD4) and thymine DNA glycosylase (TDG). In addition, uracil misincorporation into DNA can be prevented by the hydrolysis of dUTP, a precursor that can be used by DNA polymerase instead of dTTP, to dUMP and diphosphate (*Barnes & Lindahl, 2004*). This reaction is catalyzed by dUTPase, which in humans is encoded by the *DUT* gene. Although there is no direct proof of an elevated uracil level in the DNA of depressed patients, several reports indicate the presence of deficiencies in folate (vitamin $B_9$) and cobalamin (vitamin $B_{12}$) (*Papakostas et al., 2005*; *Reynolds, 2002*; *Beydoun et al., 2010*; *Kim et al., 2008*; *Young, 2007*; *Araújo et al., 2015*). These vitamins are important for nucleotides biosynthesis, and their shortage may cause the balance between dTMP and dUMP to shift towards the latter (*Reynolds, 2014*). Finally, oxidative stress may also cause the increased presence of uracil in depressed patients' DNA (*Endres et al., 2004*; *An et al., 2005*).

The evidence suggests, then, that depression may be associated with impairment of uracil metabolism, resulting in greater amounts of uracil being present in the DNA of those
with depression. Therefore, the present study examines the possible relationship between the occurrence, onset, severity or treatment efficiency of rDD and the occurrence of three SNPs located in uracil-processing genes: g.7245G>C (rs34259) of the *UNG*, c.-31A>G (rs3087404) of the *SMUG1* and g.48638795G>T (rs4775748) of the *DUT*.

## MATERIALS AND METHODS

### Study subjects and data collection

A total of 585 people were randomly selected to participate in the study: 282 patients diagnosed with DD and hospitalized at the Department of Adult Psychiatry of the Medical University of Lodz (Poland), and 303 healthy controls. No replacement sampling was performed. Detailed characteristics of the participants are presented in Table 1. All patients met the diagnostic criteria for depressive episode and recurrent depressive disorder according to WHO (*WHO, 2015*). The inclusion criteria were based on those outlined in the ICD-10 (F32.0–7.32.2, F33.0–F33.8). Prior to the start of the experiment, a standardized Composite International Diagnostic Interview (CIDI) was used to obtain a case history from each patient (*Patten, 1997*). The severity of the disease symptoms was assessed by the 21-item Hamilton Depression Rating Scale (HAM-D) (*Hamilton, 1960*). The intensity of the symptoms was measured in accordance with *Demyttenaere & De Fruyt (2003)*. Both the CIDI and HAM-D were performed by the same psychiatrist: once before the patient was included in the study, and again after antidepressant therapy with selective serotonin reuptake inhibitors (SSRIs). The exclusion criteria included the presence of concurrent somatic diseases or axis I and II disorders, other than depressive episodes, inflammatory or autoimmune disorders, central nervous system traumas, familial prevalence of mental disorders other than recurrent depressive disorders or unwillingness to give informed consent. A study group was formed of unrelated native residents of central Poland.

Participation was voluntary, and the subjects were informed of the experiment's purpose and assured of its voluntary nature. They were also informed that their personal data would be confidential. All subjects gave their written consent to participate in this study. The protocol was approved by the Bioethics Committee of the Medical University of Lodz (No. RNN/70/14/KE).

### Selection of single-nucleotide polymorphisms

Selection of the polymorphisms was performed using the public domain of the database for single nucleotide polymorphisms (dbSNP) of the National Center for Biotechnology Information, available at http://www.ncbi.nlm.nih.gov/snp (Bethesda, MD, USA). Two criteria were followed when selecting SNPs: a minor allele frequency (MAF) greater than 0.05 (population ID: HapMap-CEU), and a localization in either the coding or regulatory region of the genes.

### DNA extraction

Genomic DNA was isolated using the Blood Mini Kit (A&A Biotechnology, Gdynia, Poland) from the venous blood of depressed patients before antidepressant therapy. The purity and quantity of the DNA samples was determined by measuring their absorbance at

**Table 1  The detailed characteristic of patients which were qualified the study.**

| Depression severity (HAM-D range of scores) | Percentage of patients before treatment | Percentage of patients after treatment |
|---|---|---|
| None (0–7) | 0.41% | 68.00%[*] |
| Mild (8–16) | 13.11% | 30.67%[*] |
| Moderate (17–23) | 34.43% | 1.33%[*] |
| Severe (≥24) | 52.05% | 0%[*] |
| Mean age of patients ± SD | | 49.34 ± 10.32[#] |
| Mean age of controls ± SD | | 51.39 ± 13.37 |
| Gender (male/female) of patients | | 146/136 |
| Gender (male/female) of controls | | 155/148[&] |
| Duration of disease from the first episode | | Percentage of patients |
| 0–10 years | | 52.00% |
| 11–20 years | | 19.56% |
| 21–30 years | | 17.78% |
| 31–40 years | | 9.78% |
| ≥41 years | | 0.89% |
| Number of episodes | | Percentage of patients |
| 1 | | 13.78% |
| 2 | | 31.11% |
| 3 | | 32.00% |
| 4 | | 18.22% |
| 5 | | 4.44% |
| 6 | | 0.44% |

**Notes.**

Significance of comparisons estimated with the Yates-corrected $chi^2$ test or the Fisher exact test. [&]$P = 0.881$ vs. patients. [*]$P < 0.0001$; [#]$P = 0.225$ vs. controls.

260 nm and 280 nm. To prevent multiple freezing and thawing, the blood was aliquoted as individual samples and stored at $-20\,^{\circ}C$.

## Genotyping

TaqMan SNP Genotyping Assay (Thermo Fisher Scientific, Waltham, Massachusetts, USA) and 2X Master Mix Takyon for Probe Assay—No ROX (Eurogentec, Liège, Belgium) were used to genotype selected polymorphisms. Reactions were performed and analyzed in the Mx3005P qPCR System and MxPro QPCR Software (Agilent Technologies, Santa Clara, CA, USA).

## Statistical analysis

The collected data was analyzed in Statistica 12 (Statsoft, Tulsa, OK, USA), SigmaPlot 11.0 (Systat Software Inc., San Jose, CA, USA), Resampling Stats Add-in for Excel v.4 (Arlington, VA, USA) and StudSize3.02 (CreoStat HB, Västra Frölunda, Sweden; used for power analysis). An unconditional multiple logistic regression model was used to calculate the association between case/control and each polymorphism. The results are shown as odds ratios (ORs) with 95% confidence interval (95% CI). ORs were adjusted for gender, due to the fact that women are twice more likely to suffer from depression than men

(*Kessler, 2004*). In addition, the significant outcomes were further verified with the use of two approaches: the bootstrap-boosted multiple logistic regression (resampling with replacement, 10,000 iterations) and the cross-validated logistic regression (that corresponds to *d*-jackknife technique) (control group was the modeled class). This was intended to overcome any possible bias related to relatively low sample size. The goodness of fit of logistic regression models pointing for a significant discrimination between controls and patients was estimated with Hosmer–Lemeshow test.

The descriptive data presented in figures and tables are shown as means ±SD or medians with interquartile ranges. Normality of the studied group was assessed via the Shapiro–Wilk test, homogeneity of variance was verified with Brown–Forsythe test, and then, accordingly, either unpaired Student's t test or Mann–Whitney *U* test was used. In some bivariate and multivariate analyses we used the approach of resampling with replacement (the bootstrap-boosted versions of the tests, 10,000 iterations) to make sure that the revealed differences were not detected by a pure chance. The effect of the studied SNPs on the age of the onset of depression was evaluated in two different ways. The first approach used age as a continuous variable, while the second employed a cut-off at 35 years, an age which might be regarded as the transition between young adult to middle-aged adult.

## RESULTS

### Single-nucleotide polymorphisms of genes encoding proteins involved in the removal of uracil from DNA and the risk of recurrent depressive disorder

Table 2 presents the distribution of the genotypes and alleles of the studied polymorphisms in both depressed patients and controls. The distribution of all SNPs was in agreement with Hardy–Weinberg equilibrium. Only one of the studied SNPs modulated the risk of depression occurrence: while the G/G genotype and G allele of *UNG* g.7245G>C (rs34259) had a protective effect, the G/C genotype and C allele increased the odds ratio of the disease incidence.

### Gene-gene interactions and the risk of recurrent depression disorder

Regarding the effect of combined genotypes, it was found that the T/T-A/A-G/C (*DUT* (rs4775748), *SMUG1* c.-31A>G (rs3087404) and *UNG* (rs34259) carriers had an increased risk of depression, whereas the T/G-A/G-G/G carriers had a reduced risk (Table 3).

### Single-nucleotide polymorphisms of genes encoding proteins involved in the removal of uracil from DNA and the age of the first episode of recurrent depressive disorder

To evaluate whether the studied SNPs could have had an impact on the onset of the disease the patients were divided according to genotype and their age of onset was compared (Figs. 1A–1C). In the case of *SMUG* c.-31A>G (rs3087404), it was found that carriers of the A/A genotype underwent their first episode significantly later than the G/G carriers. Similarly, in case of *UNG* g.7245G>C (rs34259), patients with a G/G genotype had a

**Table 2  Association between the studied single-nucleotide polymorphism and depression.** Table presents a distribution of genotypes and alleles of DUT g.48346598G>T (rs4775748), SMUG1 c.-31A>G (rs3087404) and UNG g.7245G>C (rs34259) single-nucleotide polymorphisms, and OR with 95% CI in groups of patients with rDD and controls without mental disorders.

| Genotype/Allele | Control (n = 303) | | Depression (n = 282) | | Crude OR (95% CI) | p | Adjusted OR (95% CI) | p |
|---|---|---|---|---|---|---|---|---|
| | Number | Frequency | Number | Frequency | | | | |
| *DUT* g.48346598G>T (rs4775748) | | | | | | | | |
| T/T | 205 | 0.677 | 207 | 0.734 | 1.319 (0.923–1.886) | 0.128 | 1.319 (0.922–1.885) | 0.129 |
| T/G | 89 | 0.294 | 67 | 0.238 | 0.749 (0.518–1.084) | 0.126 | 0.749 (0.518–1.084) | 0.126 |
| G/G | 9 | 0.030 | 8 | 0.028 | 0.954 (0.363–2.507) | 0.924 | 0.956 (0.364–2.515) | 0.928 |
| | | | | $\chi^2 = 2.420; p = 0.298$ | | | | |
| T | 499 | 0.823 | 480 | 0.854 | 1.250 (0.915–1.706) | 0.160 | 1.249 (0.915–1.705) | 0.162 |
| G | 107 | 0.177 | 82 | 0.146 | 0.800 (0.586–1.092) | 0.160 | 0.801 (0.586–1.093) | 0.162 |
| *SMUG1* c.-31A>G (rs3087404) | | | | | | | | |
| A/A | 66 | 0.218 | 71 | 0.252 | 1.220 (0.832–1.790) | 0.309 | 1.220 (0.831–1.790) | 0.310 |
| A/G | 168 | 0.554 | 145 | 0.514 | 0.851 (0.614–1.179) | 0.331 | 0.851 (0.614–1.179) | 0.331 |
| G/G | 69 | 0.228 | 66 | 0.234 | 1.025 (0.697–1.508) | 0.899 | 1.026 (0.697–1.511) | 0.896 |
| | | | | $\chi^2 = 1.187; p = 0.552$ | | | | |
| A | 300 | 0.495 | 287 | 0.509 | 1.077 (0.843–1.358) | 0.579 | 1.070 (0.842–1.358) | 0.582 |
| G | 306 | 0.505 | 277 | 0.491 | 0.935 (0.736–1.187) | 0.579 | 0.935 (0.736–1.187) | 0.582 |
| *UNG* g.7245G>C (rs34259) | | | | | | | | |
| G/G | 178 | 0.587 | 134 | 0.475 | **0.636(0.458–0.882)*** | **0.007** | **0.636(0.459–0.882)*** | **0.007** |
| | | | | | **b0.641 (0.462–0.888)** | **0.008** | **b0.642 (0.464–0.888)** | **0.008** |
| | | | | | **cv0.636 (0.458–0.882)** | **0.007** | **cv0.636 (0.458–0.882)** | **0.007** |
| G/C | 107 | 0.353 | 127 | 0.450 | **1.480 (1.061–2.064)**** | **0.021** | **1.479 (1.059–2.065)**** | **0.021** |
| | | | | | **b1.478 (1.060–2.061)** | **0.021** | **b1.476 (1.062–2.060)** | **0.020** |
| | | | | | **cv1.480 (1.061–2.063)** | **0.021** | **cv1.479 (1.060–2.064)** | **0.021** |
| C/C | 18 | 0.059 | 21 | 0.074 | 1.274 (0.664–2.444) | 0.466 | 1.276 (0.665–2.449) | 0.464 |
| | | | | $\chi^2 = 7.401; p = 0.025$ | | | | |
| G | 463 | 0.764 | 394 | 0.702 | **0.716 (0.552–0.928)#** | **0.012** | **0.715 (0.551–0.929)#** | **0.012** |
| | | | | | **b0.627 (0.485–0.811)** | **0.0004** | **b0.631 (0.488–0.807)** | **0.0003** |
| | | | | | **cv0.710 (0.546–0.925)** | **0.011** | **cv0.711 (0.546–0.926)** | **0.011** |
| C | 143 | 0.236 | 168 | 0.298 | **1.385 (1.062–1.806)##** | **0.016** | **1.384 (1.062–1.806)##** | **0.016** |
| | | | | | **b1.561 (1.210–2.015)** | **0.001** | **b1.560 (1.211–2.015)** | **0.001** |
| | | | | | **cv1.385 (1.063–1.805)** | **0.016** | **cv1.385 (1.062–1.805)** | **0.016** |

**Notes.**

OR adjusted for sex; the superscript [b] means the bootstrap-boosted OR (resampling with replacement, 10,000 iterations); [cv] means the cross-validated OR. Statistical power $(1 - \beta)$ for significant comparisons given for the ranges: *0.782–0.791, **0.647–0.654, #0.353–0.361 and ##0.426–0.435. for all significant comparisons. $p < 0.05$ along with corresponding ORs are in bold.

greater age of onset than the heterozygotes. When the patients were stratified into the two groups based on age, it was found that the UNG g.7245G>C (rs34259) affected the disease occurrence only in patients with early onset depression, i.e., in those under 35 years of age (Table 4). However, no statistically significant difference was found between patients with early and late onset DD (Table S1).

**Table 3  Association between the combined genotypes of the studied single-nucleotide polymorphism and depression.** Table presents a distribution of combined genotypes of DUT rs4775748, SMUG rs3087404 and UNG rs34259 single-nucleotide polymorphisms, and OR with 95% CI in groups of patients with rDD and controls without mental disorders.

| Combined genotype | Control (n = 303) | | Depression (n = 282) | | Crude OR (95% CI) | p | Adjusted OR (95% CI) | p |
|---|---|---|---|---|---|---|---|---|
| | Number | Frequency | Number | Frequency | | | | |
| *DUT* (rs4775748), *SMUG1* c.-31A>G (rs3087404) and *UNG* (rs34259) | | | | | | | | |
| T/T-A/A-G/G | 37 | 0.122 | 27 | 0.096 | 0.761 (0.450–1.287) | 0.308 | 0.758 (0.448–1.283) | 0.303 |
| | | | | | **3.402 (1.423–8.133)$^{0.892}$** | **0.006** | **3.401 (1.422–8.134)$^{0.875}$** | **0.006** |
| T/T-A/A-G/C | 7 | 0.023 | 21 | 0.074 | **[b]3.411 (1.425–8.161)** | **0.006** | **[b]3.410 (1.424–8.163)** | **0.006** |
| | | | | | **[cv]3.402 (1.423–8.133)** | **0.006** | **[cv]3.401 (1.422–8.134)** | **0.006** |
| T/T-A/A-C/C | 2 | 0.007 | 4 | 0.014 | 2.165 (0.394–11.915) | 0.374 | 2.166 (0.394–11.921) | 0.374 |
| T/T-A/G-G/G | 58 | 0.191 | 52 | 0.184 | 0.955 (0.630–1.447) | 0.828 | 0.956 (0.631–1.449) | 0.833 |
| T/T-A/G-G/C | 46 | 0.152 | 49 | 0.174 | 1.175 (0.757–1.824) | 0.472 | 1.174 (0.756–1.822) | 0.476 |
| T/T-A/G-C/C | 3 | 0.010 | 5 | 0.018 | 1.805 (0.427–7.623) | 0.422 | 1.806 (0.428–7.627) | 0.421 |
| T/T-G/G-G/G | 29 | 0.096 | 24 | 0.085 | 0.879 (0.499–1.549) | 0.655 | 0.880 (0.499–1.552) | 0.659 |
| T/T-G/G-G/C | 20 | 0.066 | 19 | 0.067 | 1.022 (0.534–1.958) | 0.947 | 1.024 (0.534–1.961) | 0.944 |
| T/T-G/G-C/C | 3 | 0.010 | 6 | 0.021 | 2.174 (0.538–8.776) | 0.275 | 2.172 (0.538–8.769) | 0.276 |
| T/G-A/A-G/G | 7 | 0.023 | 7 | 0.025 | 1.076 (0.373–3.108) | 0.892 | 1.083 (0.374–3.136) | 0.883 |
| T/G-A/A-G/C | 8 | 0.026 | 10 | 0.035 | 1.356 (0.527–3.485) | 0.528 | 1.352 (0.525–3.479) | 0.532 |
| T/G-A/A-C/C | 3 | 0.010 | 1 | 0.004 | 0.356 (0.037–3.441) | 0.372 | 0.358 (0.037–3.463) | 0.375 |
| | | | | | **0.492 (0.265–0.915)$^{0.767}$** | **0.025** | **0.491 (0.264–0.914)$^{0.765}$** | **0.025** |
| T/G-A/G-G/G | 33 | 0.109 | 16 | 0.057 | **[b]0.496 (0.267–0.922)** | **0.027** | **[b]0.495 (0.266–0.921)** | **0.026** |
| | | | | | **[cv]0.492 (0.265–0.915)** | **0.025** | **[cv]0.491 (0.264–0.914)** | **0.025** |
| T/G-A/G-G/C | 19 | 0.053 | 16 | 0.057 | 0.899 (0.453–1.785) | 0.761 | 0.898 (0.452–1.784) | 0.759 |
| T/G-A/G-C/C | 4 | 0.013 | 1 | 0.004 | 0.266 (0.030–2.394) | 0.238 | 0.267 (0.030–2.401) | 0.238 |
| T/G-G/G-G/G | 9 | 0.030 | 6 | 0.021 | 0.710 (0.250–2.021) | 0.521 | 0.712 (0.250–2.028) | 0.525 |
| T/G-G/G-G/C | 4 | 0.013 | 8 | 0.028 | 2.182 (0.650–7.329) | 0.207 | 2.189 (0.652–7.356) | 0.205 |
| T/G-G/G-C/C | 2 | 0.007 | 2 | 0.007 | 1.075 (0.150–7.683) | 0.943 | 1.069 (0.149–7.651) | 0.947 |
| G/G-A/A-G/G | 2 | 0.007 | 0 | 0.000 | – | – | – | – |
| G/G-A/A-G/C | 0 | 0.000 | 0 | 0.000 | – | – | – | – |
| G/G-A/A-C/C | 0 | 0.000 | 1 | 0.004 | – | – | – | – |
| G/G-A/G-G/G | 1 | 0.003 | 2 | 0.007 | 2.157 (0.195–23.921) | 0.531 | 2.150 (0.194–23.852) | 0.533 |
| G/G-A/G-G/C | 3 | 0.010 | 4 | 0.014 | 1.439 (0.319–6.486) | 0.636 | 1.437 (0.319–6.478) | 0.637 |
| G/G-A/G-C/C | 1 | 0.003 | 0 | 0.000 | – | – | – | – |
| G/G-G/G-G/G | 2 | 0.007 | 0 | 0.000 | – | – | – | – |
| G/G-G/G-G/C | 0 | 0.000 | 0 | 0.000 | – | – | – | – |
| G/G-G/G-C/C | 0 | 0.000 | 1 | 0.004 | – | – | – | – |

**Notes.**

OR adjusted for sex; the superscript [b] means the bootstrap-boosted OR (resampling with replacement, 10,000 iterations); [cv] means the cross-validated OR. Statistical power ($1 - \beta$) for significant comparisons given in superscripts. $p < 0.05$ along with corresponding ORs are in bold.

### Single-nucleotide polymorphisms of genes encoding proteins involved in the removal of uracil from DNA and the severity of recurrent depressive disorder or treatment effectiveness

No significant difference was found between patients with different genotypes of the studied genes with regard to HAM-D (Figs. 2A–2C), different treatment efficiency or HAM-D score
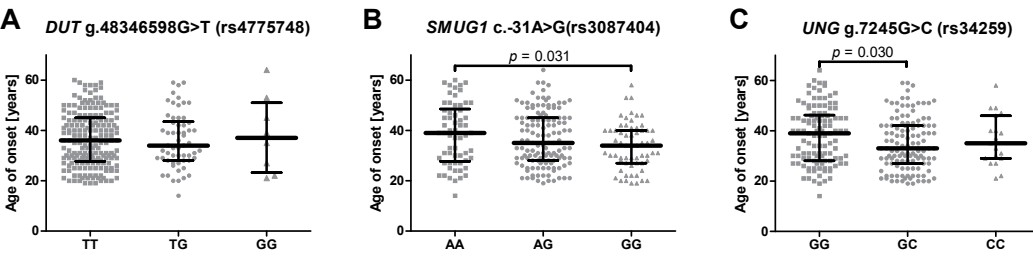

**Figure 1  Impact of single-nucleotide polymorphisms localized in uracil-processing genes on age of depression onset.** (A) *DUT* g.48346598G>T (rs4775748); (B) *SMUG1* c.-31A>G (rs3087404); (C) *UNG* g.7245G>C (rs34259). Results are presented as scatter dot plots, horizontal lines represent median, and whiskers denote interquartile range.

after treatment (Figs. 2D–2F and 2G–2I). Effectiveness of the therapy was calculated using the following formula:

$$TE = \frac{(\text{HAM-D}_0 - \text{HAM-D}_E) \times 100\%}{\text{HAM-D}_0};$$

where TE is treatment effectiveness, HAM-D$_0$ is the score obtained by the patient before therapy and HAM-D$_E$ is score achieved after therapy.

## DISCUSSION

To our best knowledge, this is the first study to demonstrate that SNPs for uracil-processing genes modulate the risk of depression. The mutagenic potential of uracil depends on its origin in DNA. Uracil is introduced into DNA either through its misincorporation by DNA polymerase during replication, or by the deamination of cytosine (*Kavli et al., 2002*), which has been estimated to occur 100–500 times per cell per day (*Frederico, Kunkel & Shaw, 1990*; *Lindahl, 1993*). In the first case, the lesion is processed by BER, and if not repaired correctly, may lead to the production of an apurinic/apyrimidinic (AP) site or a DNA strand break (*Dianov et al., 1991*; *Sousa, Krokan & Slupphaug, 2007*). In addition, the replacement of thymine by uracil in a protein binding sequence may affect the affinity of these proteins to the DNA, resulting in cytotoxic properties (*Verri et al., 1990*). In the second case, a C:G pair becomes a 100 per cent mutagenic U:G mismatch, which has been attributed to the fact that DNA polymerases cannot distinguish between U and T in the template (*Visnes et al., 2009*). Unless it is repaired, it yields C:G and U:A pairs after replication, resulting in a transition from C:G to T:A (*Sousa, Krokan & Slupphaug, 2007*).

Uracil derivatives may also possess a mutagenic character arising from oxidative stress. For example, 5-hydroxymethyluracil (5-hmeU) can be formed by the oxidation of a methyl group in thymine or deamination of 5-hydroxymethylcytosine, which is formed from 5-methylocytosine (5mC) by its oxidation: a common modification of cytosine present in human cells which is thought to be responsible for the epigenetic regulation of gene expression (*Boorstein et al., 2001*). In the light of evidence that depression may be associated with disturbances in uracil metabolism, and our own previous research showing the impairment of DNA repair in patients with depression, the present study
**Table 4  Association between the studied single-nucleotide polymorphism and the onset of depression.** Table presents a distribution of genotypes and alleles of DUT rs4775748, SMUG rs3087404 and UNG rs34259 single-nucleotide polymorphisms, and OR with 95% CI in group of patients with rDD that had their first episode before 35 years of age (marked as early onset depression) or after 35 years of age (marked as late onset depression), when compared to the control group without mental disorders.

| Geno-type/ Allele | Control (n = 303) N (Freq.) | Early onset depression (n = 127) N (Freq.) | Crude OR (95% CI) | p | Adjusted OR (95% CI) | p | Late onset depression (n = 125) N (Freq.) | Crude OR (95% CI) | p | Adjusted OR (95% CI) | p |
|---|---|---|---|---|---|---|---|---|---|---|---|
| *DUT* g.48346598G>T ( rs4775748) | | | | | | | | | | | |
| T/T | 205 (0.68) | 92 (0.72) | 1.26 (0.80–1.99) | 0.328 | 1.25 (0.79–1.30) | 0.333 | 94 (0.75) | 1.45 (0.90–2.32) | 0.123 | 1.45 (0.90–2.33) | 0.123 |
| T/G | 89 (0.29) | 31 (0.24) | 0.78 (0.48–1.25) | 0.296 | 0.77 (0.48–1.24) | 0.291 | 27 (0.22) | 0.66 (0.41–1.08) | 0.101 | 0.66 (0.41–1.09) | 0.104 |
| G/G | 9 (0.03) | 4 (0.03) | 1.06 (0.32–3.52) | 0.921 | 1.10 (0.33–3.66) | 0.875 | 4 (0.03) | 1.08 (0.33–3.57) | 0.900 | 1.05 (0.32–3.49) | 0.933 |
| $\chi^2 = 1.096; p = 0.578$ | | | | | | | $\chi^2 = 2.709; p = 0.258$ | | | | |
| T | 499 (0.82) | 215 (0.85) | 1.18 (0.79–1.76) | 0.415 | 1.17 (0.79–1.75) | 0.430 | 215 (0.86) | 1.31 (0.87–1.98) | 0.197 | 1.31 (0.87–1.98) | 0.193 |
| G | 107 (0.18) | 39 (0.15) | 0.85 (0.57–1.26) | 0.415 | 0.85 (0.57–1.27) | 0.430 | 35 (0.14) | 0.76 (0.51–1.15) | 0.197 | 0.76 (0.50–1.15) | 0.193 |
| *SMUG1* c.-31A>G ( rs3087404) | | | | | | | | | | | |
| A/A | 66 (0.22) | 26 (0.20) | 0.92 (0.55–1.54) | 0.763 | 0.91 (0.55–1.52) | 0.731 | 36 (0.29) | 1.45 (0.90–2.33) | 0.122 | 1.46 (0.91–2.35) | 0.118 |
| A/G | 168 (0.55) | 66 (0.52) | 0.87 (0.57–1.32) | 0.509 | 0.87 (0.57–1.31) | 0.502 | 63 (0.50) | 0.82 (0.54–1.24) | 0.341 | 0.82 (0.54–1.24) | 0.349 |
| G/G | 69 (0.23) | 35 (0.28) | 1.29 (0.80–2.07) | 0.291 | 1.31 (0.81–2.10) | 0.266 | 26 (0.21) | 0.89 (0.54–1.48) | 0.655 | 0.88 (0.53–1.47) | 0.629 |
| $\chi^2 = 1.118; p = 0.572$ | | | | | | | $\chi^2 = 2.401; p = 0.301$ | | | | |
| A | 300 (0.49) | 118 (0.46) | 0.87 (0.64–1.19) | 0.393 | 0.86 (0.63–1.18) | 0.360 | 135 (0.54) | 1.22 (0.89–1.66) | 0.213 | 1.22 (0.90–1.67) | 0.201 |
| G | 306 (0.50) | 136 (0.54) | 1.14 (0.84–1.56) | 0.393 | 1.16 (0.85–1.57) | 0.360 | 115 (0.46) | 0.82 (0.60–1.12) | 0.213 | 0.82 (0.60–1.11) | 0.201 |
| *UNG* g.7245G>C ( rs34259) | | | | | | | | | | | |
| | | | **0.47 (0.31–0.72)**[*] | **<0.001** | **0.47 (0.31–0.72)**[*] | **<0.001** | | | | | |
| G/G | 178 (0.59) | 51 (0.40) | [b]**0.47 (0.31–0.72)** | **<0.001** | [b]**0.47 (0.31–0.72)** | **<0.001** | 63 (0.50) | 0.71 (0.47–1.08) | 0.114 | 0.70 (0.46–1.07) | 0.102 |
| | | | [cv]**0.47 (0.31–0.72)** | **<0.001** | [cv]**0.47 (0.31–0.72)** | **0.001** | | | | | |
| | | | **1.92 (1.26–2.92)**[**] | **0.002** | **1.92 (1.27–2.91)**[**] | **0.002** | | | | | |
| G/C | 107 (0.35) | 65 (0.52) | [b]**1.91 (1.26–2.91)** | **0.002** | [b]**1.91 (1.27–2.91)** | **0.002** | 53 (0.42) | 1.35 (0.88–2.06) | 0.169 | 1.37 (0.89–2.10) | 0.147 |
| | | | [cv]**1.92 (1.26–2.92)** | **0.002** | [cv]**1.91 (1.25–2.91)** | **0.003** | | | | | |
| C/C | 18 (0.06) | 10 (0.08) | 1.35 (0.6–3.02) | 0.460 | 1.37 (0.61–3.05) | 0.445 | 9 (0.07) | 1.23 (0.54–2.81) | 0.627 | 1.21 (0.53–2.79) | 0.646 |
| $\chi^2 = 12.490; p = 0.002$ | | | | | | | $\chi^2 = 2.506; p = 0.286$ | | | | |
| | | | **0.59 (0.43–0.82)**[#] | **0.001** | **0.59 (0.42–0.82)**[#] | **0.001** | | | | | |
| G | 463 (0.76) | 167 (0.66) | [b]**0.55 (0.40–0.75)** | **0.0001** | [b]**0.54 (0.39–0.74)** | **0.0001** | 179 (0.72) | 0.78 (0.56–1.09) | 0.142 | 0.77 (0.55–1.08) | 0.133 |
| | | | [cv]**0.58 (0.42–0.81)** | **0.001** | [cv]**0.58 (0.42–0.81)** | **0.001** | | | | | |
| | | | **1.63 (1.18–2.24)**[##] | **0.003** | **1.65 (1.20–2.24)**[##] | **0.002** | | | | | |
| C | 143 (0.24) | 85 (0.34) | [b]**1.46 (1.06–2.01)** | **0.02** | [b]**1.51 (1.09–2.08)** | **0.01** | 71 (0.28) | 1.28 (0.92–1.79) | 0.142 | 1.29 (0.92–1.80) | 0.133 |
| | | | [cv]**1.66 (1.19–2.31)** | **0.003** | [cv]**1.66 (1.19–2.31)** | **0.003** | | | | | |

**Notes.**

OR adjusted for sex; the superscript [b] means the bootstrap-boosted OR (resampling with replacement, 10,000 iterations); [cv] means the cross-validated OR. Statistical power $(1 - \beta)$ for significant comparisons given for the ranges: [*]0.970–0.971, [**]0.916–0.921, [#]0.831–0.832 and [##]0.576–0.597. $P_{\text{Hosmer–Lemeshow}} = 0.152$–0.999 for all significant comparisons. $p < 0.05$ along with corresponding ORs are in bold.

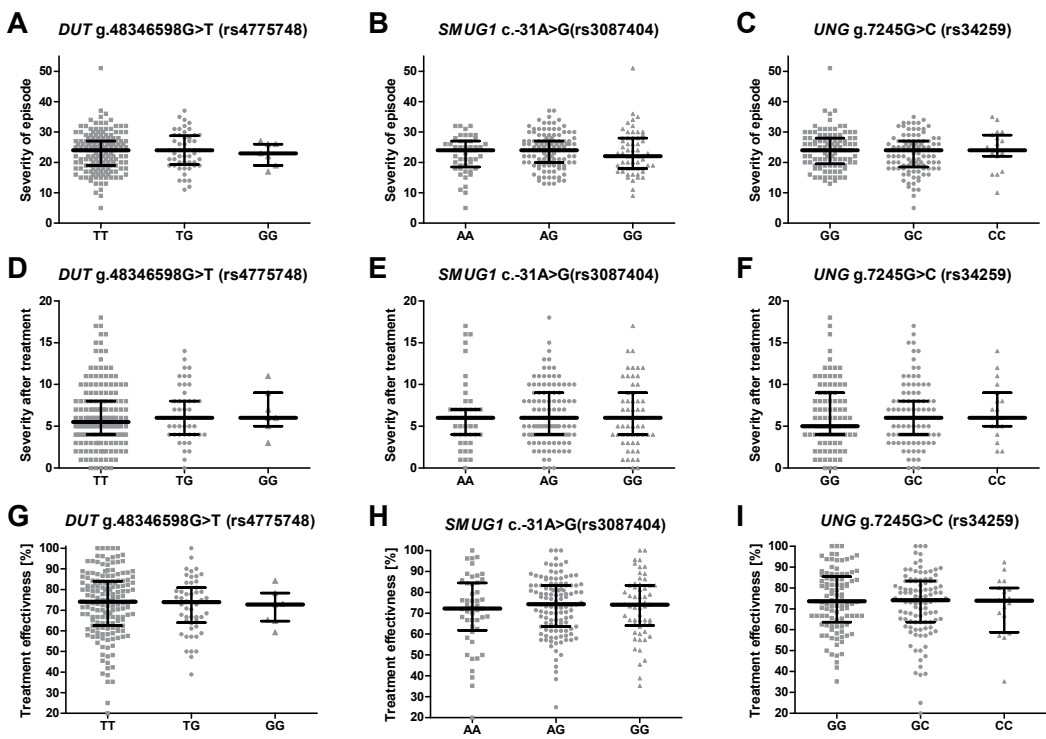

**Figure 2** **Impact of single-nucleotide polymorphisms localized in uracil-processing genes on severity of the episode before and after therapy, and on treatment effectiveness.** (A–C) Severity of current episode according to 21-item Hamilton Depression Rating Scale (HAM-D). (D–F) Severity after treatment according to HAM-D. (G–I) Treatment effectiveness expressed as percentage of HAM-D decline after treatment. Results are present as scatter dot plots, horizontal lines represent median, and whiskers denote interquartile range. In all cases $p > 0.05$.

evaluates whether the presence of SNPs within the genes of two DNA glycosylases and that of dUTPase affect the occurrence of the disease.

The first of the studied polymorphisms, *DUT* g.48346598G>T (rs4775748), is located in the downstream region of the gene encoding dUTPase, and is known to be significantly associated with blood DNA uracil concentration (*Chanson et al., 2009*); more precisely, G/G genotype carriers had lower levels of uracil in blood DNA than the other two genotypes. The authors speculate that the location of the SNP in the 3′-untranslated region causes it to modify the secondary structure of mRNA, thus influencing the stability or affinity of small RNAs to the mRNA. In contrast, the present study did not find any association between this polymorphism and the occurrence, the onset, the severity or the treatment efficiency of depression. However, the analysis of combined genotypes revealed that the heterozygote of the *DUT* SNP combined with heterozygote of *SMUG* SNP and G/G genotype of the *UNG* SNP decreased the incidence of depression, while the T/T homozygote of the *DUT* SNP combined with A/A genotype of the *SMUG* SNP and heterozygote variant of the *UNG* SNP increased it (Table 4).

The second studied SNP, *SMUG1* c.-31A>G (rs3087404), is located in a promoter region of the gene encoding SMUG1. Although its full name—single-strand-specific monofunctional uracil DNA glycosylase 1—could indicate that this glycosylase is specific to uracil in single-stranded DNA, in reality, it removes this base from both single- and double-stranded DNA in the context of U:A and U:G pairs (*Masaoka et al., 2003*; *Pettersen et al., 2007*). In addition, it recognizes uracil derivatives that have been formed by oxidation, i.e., 5-hmeU, 5-formylouracil (foU) and 5-hydoxyuracil (hmU) (*Boorstein et al., 2001*; *Masaoka et al., 2003*). SMUG1 cannot be considered only as a ''backup'' for UNG2, since its knock out results in an increased number of mutations in mouse embryonic fibroblast (MEFs) cell line (*An et al., 2005*). Moreover, SMUG1 was found to be involved in the metabolism of RNA, particularly ribosomal RNA quality control (*Jobert et al., 2013*).

The SNP was associated with age-related macular degeneration (AMD), a neurodegenerative disease causing blurred or no vision, in a Polish population (*Synowiec et al., 2014*). The polymorphism is believed to be located in the regulatory region, and so may have an impact on mRNA stability, degradation and gene expression (*Nadeau, 2002*; *Synowiec et al., 2014*). However, no such association between the SNP and the occurrence of AMD was found in a study conducted on Iranian patients (*Bonyadi et al., 2017*). Similarly, no statistically significant difference was observed between patients and controls with regard to the distribution of alleles and genotypes of this polymorphism in the present study, and the presence of the SNP did not affect the severity of depression symptoms or the effectiveness of antidepressant treatment (Figs. 2B, 2E, 2H); however, carriers with a G/G genotype reported a significantly earlier first episode than those with A/A (Fig. 1B).

The last studied polymorphism, *UNG* g.7245G>C (rs34259), is located in the downstream region of the gene encoding two forms of UNG: a mitochondrial form (UNG1) and a nuclear one (UNG2) (*Haug et al., 1998*). Both variants share a common catalytic domain, but their *N*-terminal sequences vary, resulting in different subcellular targeting (*Nilsen et al., 1997*). Like SMUG1, UNGs remove uracil from both single- and double-stranded DNA (*Kavli et al., 2002*). Moreover, UNGs recognize and excise 5-fluorouracil (5-FU) as well as oxidized pyrimidines, i.e., isodialuric acid, hmU and alloxan (*Dizdaroglu et al., 1996*; *Krokan, Standal & Slupphaug, 1997*). Although *UNG*-deficient mice are viable, they accumulate uracil in their DNA, and the mutation rate of their cells is five-times greater (*Andersen et al., 2005b*; *Nilsen et al., 2000*; *An et al., 2005*). Interestingly, they develop lymphoid hyperplasia and B-cell lymphomas, indicating that the lack of UNG disrupts the adaptive immunity (*Andersen et al., 2005a*; *Nilsen et al., 2003*).

Analogically to *DUT* g.48346598G>T (rs4775748), the *UNG* SNP (rs34259) has also been found to modulate the amount of uracil in blood DNA (*Chanson et al., 2009*). However, while the effect of the *DUT* SNP best fitted a recessive model, i.e., the heterozygotes did not differ from homozygous wild-type individuals, the *UNG* SNP displayed a more additive effect, with G/G genotype carriers having the lowest concentration of uracil in blood DNA, the heterozygotes a moderate level and the C/C genotype carriers with the highest. Chanson et al. speculate that variant C may modify the secondary structure of the mRNA, thus decreasing the level of UNG synthesis. Accordingly, in the present study, the G/G genotype and allele G decreased the risk of depression occurrence, while the G/C genotype

and allele C increased it; however, no association was found between the occurrence of C/C genotype and the disease, which may be due to the small number of individuals carrying this genotype. Nevertheless, this polymorphism may also affect the onset of depression: G/G carriers demonstrated significantly later onset of the disease than the heterozygotes (Fig. 1C). In addition, the polymorphism was found to affect the occurrence of depression only in patients that had their first episode before 35 years of age, while no association was found in patients with the first episode at or after 35 years of age (Table 3). This could indicate that the G/G genotype has a protective effect against the occurrence of depression in early life. As with the other SNPs included in the study, no evidence was found that the *UNG* SNP could affect the severity of the symptoms and the treatment outcome (Figs. 2C, 2F, 2I).

Our findings are especially interesting in the light of our earlier papers reporting impairment of DNA damage repair and those by other teams suggesting disturbances of uracil metabolism in the course of depression. As mentioned in the Introduction, folate and cobalamin deficiencies may play a role in the etiology of the disease (*Papakostas et al., 2005*; *Reynolds, 2002*; *Beydoun et al., 2010*; *Kim et al., 2008*; *Young, 2007*; *Araújo et al., 2015*). In the brain, both of these vitamins play a crucial role as cofactors in the synthesis of neurotransmitters, hormones, myelin and membrane phospholipids, as well as in epigenetic regulation (*Hughes et al., 2013*; *Reynolds, 2006*). Interestingly, they are also important for the biosynthesis of nuclear acids, particularly the formation of dTMP from dUMP. Thus, a deficiency of the vitamins causes a shifting of the balance between these two nucleoside monophosphates towards dUMP resulting in the incorporation of uracil into DNA (*Reynolds, 2014*). This, together with the impairment of DNA damage repair discovered by our team, that can be, at least partly, attributed to the presence of specific SNP variants, may lead to the introduction of AP sites or DNA strand breaks and apoptosis of neurons, especially in the hippocampus (*Young, 2007*; *Czarny et al., 2017a*). In addition, folate deficiencies are also known to affect mtDNA, causing *inter alia* an increased number of deletions and thus the induction of oxidative stress (*Chou & Huang, 2009*; *Chou, Yu & Huang, 2007*; *Kronenberg et al., 2011*). This suggests that the impairment of uracil metabolism may contribute to the dysfunction of mitochondria and the increased level of oxidative stress observed in depression (*Araújo et al., 2015*). Our results indicating that the *UNG* variant is associated with the decreased level of uracil in blood DNA, and a lower risk of disease development, further support this hypothesis. On the other hand, the results suggest that the studied SNPs did not influence neither the effectiveness of antidepressant therapy nor the severity of depression before and after treatment. In contrast, it was proven that folate deficiency may be associated with a bad response to therapy (*Papakostas et al., 2005*; *Reynolds, 2002*). Thus, we speculate that the impairment of uracil removal from DNA may be an adjunctive factor to the shortage of folate, which speeds up the development of depression, but affects neither the severity nor the treatment of the disease. However, those assumptions must be taken with caution, since the level of uracil or folate was not measured.

The study does have some limitations. While the sample size is relatively small, similar sample sizes have been used in recently published studies concerning polymorphic

variability (*Jung et al., 2017*; *Peitl, Štefanović & Karlović, 2017*). Moreover, due to the ethnic homogeneity of the studied population, results obtained by our team cannot be extrapolated to the world population. Furthermore, as no expression analysis was performed of the studied genes on the mRNA or on the protein level, it is not certain that the presence of the studied SNPs causes changes in the expression of those genes in patients with depression.

## CONCLUSION

The functional polymorphism located in *UNG* modulated the occurrence and the onset of depression, and the *SMUG1* SNP affected the time of the first episode. Our results, particularly the evidence of the protective effect of the *UNG* SNP variant in patients with the early-onset depression and the fact that this variant is associated with a low level of uracil, suggest that disturbances of uracil removal from DNA may be an additional factor, apart from folate shortage, to hasten the development of the disease. This study further supports the hypothesis that DNA damage and the impairment of DNA repair are important for the pathophysiology of depression. Further research, including the analysis of levels of uracil incorporation into DNA of depressed patients, is needed to elucidate the potential casual-effect mechanisms.

### Funding
This work was supported by the Medical University of Lodz, Poland [Research Program No. 502-03/6-086-01/502-64-104]. The funders had no role in study design, data collection and analysis, decision to publish, or preparation of the manuscript.

### Grant Disclosures
The following grant information was disclosed by the authors:
Medical University of Lodz, Poland: 502-03/6-086-01/502-64-104.

### Competing Interests
The authors declare there are no competing interests.

### Author Contributions

- Piotr Czarny conceived and designed the experiments, performed the experiments, analyzed the data, contributed reagents/materials/analysis tools, prepared figures and/or tables, authored or reviewed drafts of the paper, approved the final draft.
- Paulina Wigner performed the experiments, analyzed the data, prepared figures and/or tables.
- Justyna Strycharz and Ewa Swiderska performed the experiments, authored or reviewed drafts of the paper, approved the final draft.
- Cezary Watala analyzed the data, contributed reagents/materials/analysis tools, authored or reviewed drafts of the paper, approved the final draft.

- Ewelina Synowiec performed the experiments.
- Piotr Galecki conceived and designed the experiments, contributed reagents/materials/analysis tools, authored or reviewed drafts of the paper, approved the final draft, diagnosis of the patients.
- Monika Talarowska performed the experiments, analyzed the data, authored or reviewed drafts of the paper, diagnosis of the patients.
- Janusz Szemraj conceived and designed the experiments, analyzed the data, contributed reagents/materials/analysis tools, authored or reviewed drafts of the paper, approved the final draft.
- Kuan-Pin Su conceived and designed the experiments, analyzed the data, authored or reviewed drafts of the paper, approved the final draft.
- Tomasz Sliwinski conceived and designed the experiments, analyzed the data, contributed reagents/materials/analysis tools, authored or reviewed drafts of the paper, approved the final draft.

## Human Ethics

The following information was supplied relating to ethical approvals (i.e., approving body and any reference numbers):

This study was approved by the Bioethics Committee of the Medical University of Lodz (No. RNN/70/14/KE).

## Data Availability

The raw data are provided in a Supplemental File.

## Supplemental Information

Supplemental information for this article can be found online at http://dx.doi.org/10.7717/peerj.5116#supplemental-information.

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
