# Peer review of "Single-nucleotide polymorphisms of uracil-processing genes affect the occurrence and the onset of recurrent depressive disorder"

_PeerJ, doi:10.7717/peerj.5116_

## Round 0.1 · original submission · Major Revisions

Your manuscript was revised by 3 reviewers. As you can see, two of them recommended publication, but Reviewer 2 recommended rejection. The major concerning was about the sample size. There are also other concerns that you have to consider, before a final decision about the acceptance of your manuscript can be reached.

Reviewer 1 ·

Basic reporting

The English language should be improved by a linguist, because of some errors, i.e.,
43. patients suffers – suffer
57 an exposure –exposure
63 This leads - It is unclear what This refers to
66 for a removal-for removal
79 shifting of balance-shifting of the balance

and the like.

Experimental design

No comment

Validity of the findings

The strong side of the work is well documented, in the introduction, the occurrence of the oxidative DNA damage in depressed patients, providing the foundation for the research undertaken by the authors. The genes and their polymorphisms were selected adequately to the presented hypothesis, although the list has not been exhausted.

Additional comments

The weakness of the work is the lack of consideration of the power analysis when designing the study. It is not known whether the number of patients tested is sufficient for correct inference.

It is not clear to the reviewer why the authors divided patients due to the age of onset of the disease before and after the age of 35 - explanation is necessary.

As a researcher, I would suggest to authors to supplement research and analyze the expression of the analyzed genes because the space-time shift and numerous transcriptional and post-translational modifications can significantly alter the final effect of the genes studied.

Reviewer 2 ·

Basic reporting

Not appropriate.
In Discussion section, genomic mechanisms beyond the current study are mentioned, which seem just specuration beyond the current findings.

Experimental design

Generally excellent, but small sample size.
The author report statistical power based on the previous studies.

Validity of the findings

Not enough.
The authors show clinical usefulness of the current findings.

Additional comments

This study assessed the associations between SNPs of uracil-processing genes and occurrence or onset of recurrent depressive disorder as well as treatment effectiveness. Although the authors found some significant associations between studied SNPs and depression, they failed to find the influence of those SNPs on effectiveness antidepressant therapy nor the severity of depression before and after the treatment.
(1) Although study protocol using case-control design is advantage of the current study, the sample size seems to be too small to detect any potential associations between those SNPs and characteristics of depressive disorder (usually 5000 controls and 5000 cases are required for this kind of analysis of susceptibility genes). Many non-significant associations observed in the current study may due to insufficient sample size. The authors should report statistical power of the current study based on the previous findings regarding this issue.
(2) In addition, the calculated effect size is not so large (odds ratio ranged from 0.5-1.7) as to explain the clinical significance of these SNPs on the disorder. What can clinicians do next step based on the current findings? What is the clinical significance of these findings? Without these answers, the valuable case-control design is spoilt. Relatively large effect size is observed between the combination of three genes and depression (in Table4, T/T-A/A-G/C; adjusted odds ratio =3.4), but this was calculated by only 21 patients and 7 controls, resulting in wide range of confidence interval.
(3) Most basically, since the authors did not show which genotype is the referent, readers could not understand what the presented odds ratios mean, i.e., how many times the studied SNPs are risky or protective in comparison with WHAT GENOTYPE.
(4) In my opinion, this kind of study using common SNPs should focus much on treatment effectiveness to clarify the clinical usefulness. However, unfortunately, the authors failed to find any clinical effectiveness related to the studied SNPs, which may be, at least in part, due to the small sample size as mentioned above.

Reviewer 3 ·

Basic reporting

no comment.

Experimental design

no comment.

Validity of the findings

no comment.

Additional comments

This paper investigated the influences of three SNPs localized in genes encoding uracil-processing proteins on occurrence and the onset of depression. Although the onset of depression can be determined only by rs2337395, their findings were very important and interesting for further studies.

It would be interested to see whether rs2337395 showed similar significant influence after adjusted for age in addition to sex in the logistic analyses. Also, it would be nice to see actual median and interquartile data in the figures.

---

## Round 0.2 · accepted · Accept

Thank you for revising the original manuscript.

# Reviewer 1 ·

Basic reporting

no comment

Experimental design

no comment

Validity of the findings

no comment

Additional comments

The authors took into account all my comments and recommendations, they supplemented the calculations and explanations. I recommend the manuscript for publication.

Reviewer 3 ·

Basic reporting

no comment

Experimental design

no comment

Validity of the findings

no comment

Additional comments

no comment